# How does framing influence preference for multiple solutions to societal problems?

**James Shyan-Tau Wu**[1]*, **Claire Kremen**[1,2,3], **Jiaying Zhao**[1,4]

**1** Institute for Resources, Environment and Sustainability, University of British Columbia, Vancouver, British Columbia, Canada, **2** Department of Zoology, University of British Columbia, Vancouver, British Columbia, Canada, **3** Biodiversity Research Centre, University of British Columbia, Vancouver, British Columbia, Canada, **4** Department of Psychology, University of British Columbia, Vancouver, British Columbia, Canada

\* james.shyantau.wu@ubc.ca

**Data Availability Statement:** The dataset and the R code for this study are available on OSF: https://osf.io/k4ud5/.

**Funding:** The author(s) received no specific funding for this work.

## Abstract

Solutions to environmental and social problems are often framed in dichotomous ways, which can be counterproductive. Instead, multiple solutions are often needed to fully address these problems. Here we examine how framing influences people's preference for multiple solutions. In a pre-registered experiment, participants (N = 1,432) were randomly assigned to one of four framing conditions. In the first three conditions, participants were presented with a series of eight problems, each framed with multiple causes, multiple impacts, or multiple solutions to the problem. The control condition did not present any framing information. Participants indicated their preferred solution, perceived severity and urgency of the problem, and their dichotomous thinking tendency. Pre-registered analyses showed that none of the three frames had a significant impact on preference for multiple solutions, perceived severity, perceived urgency, or dichotomous thinking. However, exploratory analyses showed that perceived severity and urgency of the problem were positively correlated with people's preference for multiple solutions, while dichotomous thinking was negatively correlated. These findings showed no demonstrable impact of framing on multi-solution preference. Future interventions should focus on addressing perceived severity and urgency, or decreasing dichotomous thinking to encourage people to adopt multiple solutions to address complex environmental and social problems.

## Introduction

Solutions to environmental and social problems (e.g., climate change, land management) are often framed in a dichotomous fashion [1–3]. For example, solutions to climate change are framed as either government action, such as enacting environmental regulations and implementing carbon taxes, or individual action, such as driving less, flying less, and eating less meat [4–6]. In reality, both government and individual actions are generally needed and neither should come to replace the other [7]. The dichotomous frame of solutions is partly driven by dichotomous thinking, that is, the tendency to categorize things, people, and events into two contrasting categories [8–10]. Dichotomous thinking is pervasive in all sectors of society:

**Competing interests:** The authors have declared that no competing interests exist.

the two dominant political parties corresponding to left and right-wing political ideologies in many countries [11], the characterization of cultures as either Eastern or Western [12], and the scientific tendency to classify results as significant or not, based on an arbitrary threshold, the p-value [13]. Dichotomous thinking simplifies the complexities of reality (many shades of grey) into two discrete opposing categories (black versus white), which reduces cognitive load and allows humans to make decisions more efficiently [10].

However, in the context of solving important societal problems, dichotomous thinking has considerable downsides. Taking climate change mitigation as an example, its division into government and individual actions can be counterproductive, since effective climate action requires cooperation among all sectors of society [14]. Another example is the encroachment of agriculture on natural lands which subsequently leads to biodiversity loss. Two contrasting actions have been proposed: land-sparing, which is limiting highly intensive agriculture across a small area of land to spare land for nature, and land-sharing, which is practicing wildlife-friendly, less-intensive agriculture across a large area of land [15, 16]. However, a growing body of evidence suggests that both approaches are needed, and relying on just one or the other can lead to suboptimal outcomes for both agricultural productivity and biodiversity [2, 17, 18]. These examples not only highlight the flaws of dichotomous thinking, but also the importance of a multi-solution approach to address complex problems.

Thus, there is a need to develop framing interventions to reduce dichotomous thinking and promote a multi-solution approach. However, dichotomous thinking has been primarily studied as a personality trait predictive of mental and personality disorders [8, 19, 20], rather than in the context of environmental and social problems, or as an intervention point for behavioral change [21]. This said, previous research has examined framing as a behavioral intervention to promote the awareness of and action on important societal problems [22, 23]. Many past framing interventions aimed to reduce the psychological distance between the individual and the problem, with the idea being if a problem becomes more personally relatable, there is a higher chance that the individual will act to address it [22, 24]. For example, to elicit individual awareness and action on climate change, some studies use local frames which emphasize the impact of local extreme weather resulting from climate change instead of global increases in temperature and sea levels [25, 26]. Other studies frame climate change as a personal health problem, highlighting its relationship with worsening air quality and the spread of infectious diseases, instead of presenting it as an environmental problem [27, 28]. Aside from making a problem more relatable, these framings could also help people see a broader picture and understand different sides of a problem [29, 30], which is useful in reducing dichotomous thinking.

Knowing this, multi-cause, multi-impact, or multi-solution framing might be employed encourage a multi-solution approach. The multi-cause frame highlights that complex problems are the result of multiple causes, which promotes a broader understanding of a problem [31] and reveals the consequences of personal choices [32]. Understanding these subtleties can allow people to understand how each aspect of a problem needs to addressed using a unique solution [31, 33]. The multi-impact frame emphasizes how there are multiple impacts caused by a problem, which could increase the personal relatability of a problem, since at least one of the impacts might be relevant to a person's daily life [34, 35]. Furthermore, being exposed to information on multiple impacts could increase people's concern for a problem, which makes them more likely to support a wide range of measures to tackle a problem [35–37]. Lastly, the multi-solution frame explicitly points out multiple solutions to a complex problem. This frame could address people's lack of knowledge on solutions, as well as outline the trade-offs between different solutions, underscoring how single solutions are inadequate [38].

The goal of the current study is to examine how framing influences people's preference for multiple solutions to societal problems and whether it changes people's tendencies towards

dichotomous thinking. Specifically, we conducted a pre-registered experiment where we designed and tested three frames against a control condition: a multi-solution frame that explicitly calls for the need to use multiple solutions for a problem, a multi-cause frame that presents the multiple causes of a problem, and a multi-impact frame that presents the multiple negative impacts of a problem. In the experiment, participants were randomly assigned to one of the four conditions (control, multi-cause, multi-impact or multi-solution frame) and read descriptions of eight environmental and social problems presented in a random order (see S1 File). The eight problems included climate change [4], crop yield and biodiversity [2], food waste [39], plastic pollution [40], homelessness [41], police reform [42], public education [43], and early pandemic response [44]. The problems were chosen to represent a diverse and timely assortment of societal problems. Due to the severity, pressing urgency, and complexity of these problems, a multi-solution approach is necessary in each case. After reading each problem, participants rated the perceived severity and perceived urgency of the problem, and indicated their preferred solution to address the problem among four options. The four options included two distinct solutions that were usually portrayed in dichotomous ways, a multi-solution option, and neither of the first two solutions. Participants' choice of the multi-solution option was the primary measure of interest. At the end of the experiment, participants completed the Dichotomous Thinking Inventory which was a validated questionnaire to measure their level of dichotomous thinking [45].

## Materials and methods

### Pre-registration and ethics approval

The experiment was pre-registered on the Open Science Framework (OSF): https://osf.io/gx64r. The study was approved by the University of British Columbia's Behavioral Research Ethics Board (ID: H22-01832). All participants provided written informed consent prior to their participation in the study. All statistical analyses were conducted in R Version 4.2.1. The dataset and the R code are available on OSF: https://osf.io/k4ud5/.

### Participants

Data collection for this study occurred in August 2022. A total of 1,630 participants completed the study without failing the attention check. After applying our pre-registered exclusion criteria, a final sample of 1,432 participants were included in the analysis. This number slightly exceeded the required sample size of 1,424 from an a priori power analysis (conducted in G*Power Version 3.1.9.6) for a one-way ANOVA, where we assumed a minimum effect size of d = 0.1, alpha = 0.05, and power = 0.9. This study was conducted on Amazon Mechanical Turk (MTurk), an online platform where participants received payment for completing surveys and other tasks. In our study, we required MTurk participants to be based in the U.S. and have an approval rate greater than or equal to 95%. A majority of our participants were female (54.49%), white (81.18%), and highly educated (73.74% had a bachelor's degree or above). Participants were on average middle-age (M = 38.53, SD = 12.21), liberal leaning (45.57% liberal, 35.81% conservative), and had a median annual personal income of US$31,754. There were no significant differences in any of the demographic variables between conditions.

### Procedures

We used a randomized between-subjects design with four conditions (control, multi-cause, multi-impact, and multi-solution) to examine the effect of the framing on preference for the

multi-solution option, perceived severity, perceived urgency, and level of dichotomous thinking. Participants were randomly assigned to one of the four conditions. In all four conditions, participants were presented with information on eight societal problems (climate change, crop yield and biodiversity, food waste, plastic pollution, homelessness, police reform, public education, early pandemic response) in a random order (see S1 File). Because this study involved U. S. participants, these problems were presented and tailored to a U.S. context. In the control condition (N = 356), participants were presented with a short paragraph presenting an overview of a given problem in neutral language with several bullet points presenting multiple facts about the problem. In the multi-cause condition (N = 358), participants were presented with the same paragraph with bullet points describing the multiple causes of the problem. In the multi-impact condition (N = 361), participants were presented with the same paragraph with bullet points describing the multiple impacts of the problem. In the multi-solution condition (N = 357), participants were presented with the same paragraph with bullet points describing the multiple solutions to the problem. For each problem, the description and bullet points contained the same number of words across the four conditions. To make sure participants read each problem, we added a manipulation check where participants had to arrange the bullet points in the order they were presented. Participants could only proceed when they had correctly ordered the bullet points. After reading the problem, participants indicated the perceived severity and perceived urgency of the problem, and chose their preferred solution among four options (see S1 File). The four options included two distinct solutions that were framed in dichotomous ways, a multi-solution option, and neither of the first two solutions. At the end of the experiment, participants completed the Dichotomous Thinking Inventory which was a validated questionnaire to measure their level of dichotomous thinking [42], and answered some demographic questions.

## Measures

**Perceived severity.** Perceived severity, a measure of how severe participants think a problem is, was measured on a 7-point Likert scale (1 = not severe at all, 2 = slightly severe, 3 = somewhat severe, 4 = moderately severe, 5 = considerably severe, 6 = very severe, 7 = extremely severe). Average perceived severity was obtained by averaging the perceived severity of the eight problems for each participant.

**Perceived urgency.** Perceived urgency, a measure of how urgent participants think a problem is, was measured on a 7-point Likert scale (1 = not urgent at all, 2 = slightly urgent, 3 = somewhat urgent, 4 = moderately urgent, 5 = considerably urgent, 6 = very urgent, 7 = extremely urgent). Average perceived urgency was obtained by averaging the perceived urgency of the eight problems for each participant.

**Preferred solution.** Preferred solution is the primary pre-registered measure and is participant's choice of solution to a given problem out of the four options. Option 1 and Option 2 were binary opposing solutions (e.g., government action and individual action for climate change). Option 3 is the multi-solution option, worded as "Multiple solutions are needed in addition to 1 and 2". Option 4 is neither, which is worded as "Neither 1 nor 2". For each participant, we calculated the total number of multi-solution options chosen out of the eight problems. The total number of multi-solution options chosen is referred to as preference for multiple solutions in the analysis.

**Level of dichotomous thinking.** The level of dichotomous thinking was measured using the Dichotomous Thinking Inventory [42], which is a questionnaire consisting of 15 statements (see S2 File). Participants rated how much they agreed with each item on a 6-point Likert scale (1 = strongly disagree, 6 = strongly agree). The level of dichotomous thinking was

obtained by averaging the scores of the 15 ratings. The Dichotomous Thinking Inventory was completed after participants have responded to the eight problems, and is therefore a measure of a person's tendency towards dichotomous thinking, instead of serving as the baseline level of dichotomous thinking.

**Demographic covariates.** We collected demographic information on gender, age, ethnicity, political orientation, education level, annual household income, and household size as demographic variables (see S3 File). Annual personal income was calculated by dividing annual household income by the square root of household size.

# Results

## Pre-registered analyses

To examine the first pre-registered hypothesis (participants who are presented with the multi-cause, multi-impact, and multi-solution frame will be more likely to choose the multi-solution option compared to the control condition), we conducted a Kruskal-Wallis test to compare the number of multi-solution choices between the four conditions (Fig 1a). This is because our data violated assumptions of normality (Shapiro-Wilk: $W = 0.90$, $p < .001$), so a Kruskal-Wallis test was used instead of a one-way ANOVA. The Kruskal-Wallis test showed no significant difference between conditions ($\chi^2(3) = 6.21$, $p = .10$), which means that the hypothesis was not supported. Per our pre-registration, we also conducted a one-way ANCOVA to examine the effect of demographic covariates (gender, age, ethnicity, political orientation, education level, and annual personal income); in line with the Kruskal-Wallis test, there was no significant main effect of condition ($F(3,1428) = 1.71$, $p = .16$), while all six demographic covariates were significant (see S1 Table in S4 File). Fig 1a shows the number for multiple solution choices in each condition (control: $M = 4.60$, $SE = .15$; multi-cause: $M = 4.20$, $SE = .15$; multi-impact: $M = 4.27$, $SE = .15$; multi-solution: $M = 4.58$, $SE = .15$).

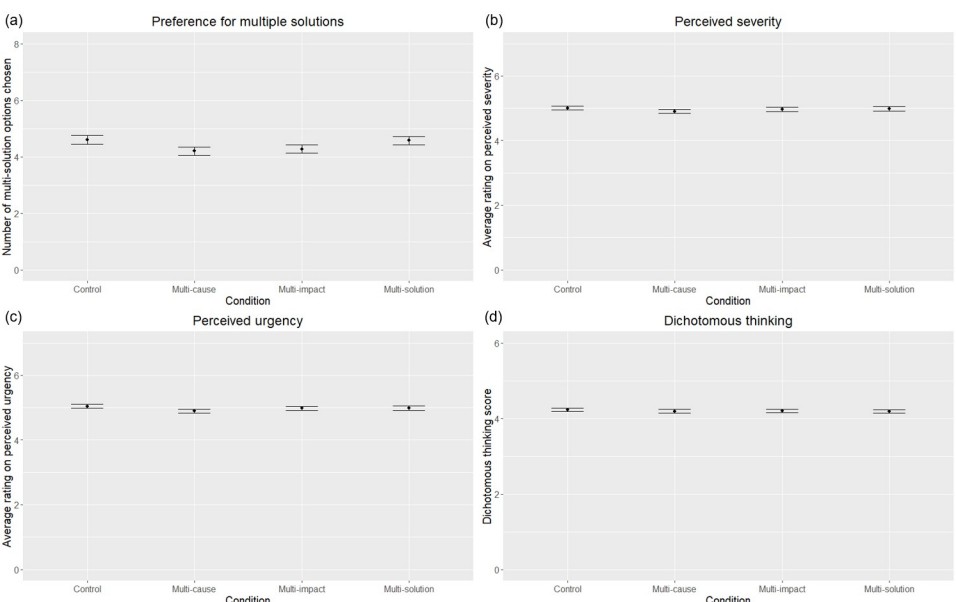

**Fig 1.** (a) Preference for multi-solutions, which is defined as the number of multi-solution option chosen out of the eight problems in each condition. (b) Average perceived severity in each condition. (c) Average perceived urgency in each condition. (d) Dichotomous thinking in each condition. Error bars represent ±1 standard error.

To examine the second pre-registered hypothesis (perceived severity and urgency will be higher for participants who are presented with the multi-impact frame, followed by the multi-cause and multi-solution frame, and finally the control condition), we conducted a Kruskal-Wallis test to compare the perceived severity between the four conditions (Fig 1b), due to violation of assumptions of normality (Shapiro-Wilk: $W = 0.98$, $p < .001$). The Kruskal-Wallis test showed no significant difference between conditions ($\chi^2(3) = 1.18$, $p = .76$), which means that the hypothesis was not supported. Per our pre-registration, we also conducted a one-way ANCOVA to examine the effect of demographic covariates; there was no significant main effect of condition ($F(3,1428) = 0.42$, $p = .74$), while gender, ethnicity, and political orientation were significant covariates (see S2 Table in S4 File). Fig 1b shows the average perceived severity of the eight problems in each condition (control: $M = 5.00$, $SE = .06$; multi-cause: $M = 4.91$, $SE = .07$; multi-impact: $M = 4.97$, $SE = .06$; multi-solution: $M = 4.99$, $SE = .06$).

We also conducted a Kruskal-Wallis test to compare the perceived urgency between the four conditions (Fig 1c), due to violation of assumptions of normality (Shapiro-Wilk: $W = 0.98$, $p < .001$). The Kruskal-Wallis test showed no significant difference between conditions ($\chi^2(3) = 2.83$, $p = .42$), which means that the hypothesis was not supported. Per our pre-registration, we also conducted a one-way ANCOVA to examine the effect of demographic covariates; there was no significant main effect of condition ($F(3,1428) = 0.91$, $p = .44$), while gender, ethnicity, and political orientation were significant covariates (see S3 Table in S4 File). Fig 1c shows the average perceived urgency of the eight problems in each condition (control: $M = 5.05$, $SE = .06$; multi-cause: $M = 4.89$, $SE = .06$; multi-impact: $M = 4.98$, $SE = .06$; multi-solution: $M = 4.98$, $SE = .06$).

Overall, the pre-registered analyses show that there was no significant difference in the preference for multiple solutions, perceived severity, and perceived urgency between conditions. This suggests that the framing of multiple causes, impacts, or solutions had little influence on the multi-solution preference.

## Exploratory analyses

As exploratory analyses, we examined whether framing had any impact on dichotomous thinking (Fig 1d). A Kruskal-Wallis test revealed no significant difference between conditions ($\chi^2(3) = 0.47$, $p = .93$). Fig 1d shows the average level of dichotomous thinking in each condition (control: $M = 4.23$, $SE = .05$; multi-cause: $M = 4.19$, $SE = .05$; multi-impact: $M = 4.20$, $SE = .05$; multi-solution: $M = 4.18$, $SE = .05$).

We further examined the proportion of participants who chose each option for each of the eight problems (see S1 Fig in S4 File). For almost every problem, a majority of participants (between 50% and 60%) chose the multi-solution option, while the rest were split between one of the two binary options. These patterns were consistent across all four conditions, and showed that the participants generally preferred multiple solutions. We conducted chi-squared tests for the eight problems to see whether the percentage of participants that chose the multi-solution option differed between the four conditions. The results showed that there was no significant difference between conditions for any problem (see S4 Table in S4 File). In addition, to see whether the participants became more or less likely to choose the multi-solution option as they proceeded through the study, we examined the percentage of participants that choose each option in temporal sequence (i.e., item number 1 represents the first problem a participant is presented). We found that the participants did not seem to gravitate toward certain options as time went on (see S2 Fig in S4 File).

To examine what predicts the preference for multiple solutions, we ran a series of correlations between perceived severity, perceived urgency, dichotomous thinking and preference for

multiple solutions. The correlations (see S5 Table in S4 File) showed that preference for multiple solutions was positively correlated with perceived severity (control: $r = .25$, $p < .001$; multi-cause: $r = .21$, $p < .001$; multi-impact: $r = .16$, $p < .001$; multi-solution: $r = .29$, $p < .001$), positively correlated with perceived urgency (control: $r = .28$, $p < .001$; multi-cause: $r = .24$, $p < .001$; multi-impact: $r = .18$, $p < .001$; multi-solution: $r = .29$, $p < .001$), but negatively correlated with dichotomous thinking (control: $r = -.10$, $p = .001$; multi-cause: $r = -.12$, $p = .002$; multi-impact: $r = -.13$, $p < .001$; multi-solution: $r = -.12$, $p = .002$).

To understand how demographic variables were associated with the preference for multiple solutions, we ran a series of correlations between demographic variables and preference for multiple solutions. The correlations (see S5 Table in S4 File; see also S3 Fig in S4 File) showed that there was a negative correlation between political orientation and preference for multiple solutions in most conditions (control, $r = -.16$, $p < .001$; multi-impact, $r = -.15$, $p < .001$; multi-solution, $r = -.1$, $p = .014$), meaning that the more conservative a participant was, the less likely they were to choose multi-solution options. There was also a negative correlation between education level and preference for multiple solutions in most conditions (multi-cause condition, $r = -.11$, $p = .01$; multi-impact condition, $r = -.17$, $p < .001$; multi-solution condition, $r = -.14$, $p < .001$), meaning that the more highly educated a participant was, the less likely they were to choose multi-solution options.

The final exploratory analysis was a multiple regression (GLM) with binomial distribution. In this GLM, the eight problems were computed as eight trials, where choosing a multi-solution option counts as a success, while all other choices (binary options, neither) count as a failure. The purpose of this GLM is to run a multiple regression with all variables measured in this study, as well as providing a more robust alternative to ANCOVA. The results (see S6 Table in S4 File) are generally consistent with the aforementioned analyses.

## Discussion

The current study examined the influence of framing on people's preference for a multi-solution approach to societal problems. We found framing the multiple causes, impacts, or solutions of the problem had little impact on the preference for multiple solutions, perceived severity, or perceived urgency of the problem. This null result is largely driven by the high baseline of preference for multiple solutions, as can be seen in the majority of participants that preferred the multi-solution option in the control condition (Fig 1). There are several possible reasons behind this high baseline. First, participants might have thought that employing more solutions is more effective when addressing a complex problem [46, 47]. This "more is better" mindset can be further compounded by a person's concern for a problem (i.e., more solutions feel more assuring), and the lack of resource constraints (i.e., without financial constraints it feels intuitive to employ all possible solutions), which can further encourage people to choose the multi-solution option by default [48, 49]. Second, many participants might have preferred not to take a side in these problems, a lot of which are contentious and potentially damaging to interpersonal relationships [50, 51]. Under such a mindset, choosing the multi-solution option might have allowed them to feel less divisive. Third, many participants might have not known enough about the problems or solutions, thus defaulting to the middle option out of uncertainty [52]. Lastly, high preference for multiple solutions might be partially attributed to survey design. Specifically, choosing the multi-solution option might have required less cognitive effort due to the way the options are worded. Because the multi-solution option is shorter than the two binary solutions (see S1 File), it is possible that after a quick reading, participants who considered the two binary solutions somewhat plausible would be inclined to choose the multi-solution option, which is shorter, instead of carefully considering the two binary

solutions, which requires more effort. Furthermore, the text for the multi-solution option was identical throughout the eight problems in the survey (see S1 File), which could mean it required less effort to choose the multi-solution option by default, instead of considering the set binary solutions which are unique to each problem. In addition, on a more fundamental level, the null results in the preference for multiple solutions, perceived severity, or perceived urgency of the problem could also be attributed to the small effect size of framing interventions aimed at reducing psychological distance. In fact, according to one recent study, there is weak empirical support for the effectiveness of such interventions [53].

An exploratory finding was that higher perceived severity and urgency was correlated with a stronger preference for multiple solutions, and higher dichotomous thinking was correlated with a weaker preference for multiple solutions. These patterns were consistent across all conditions. This suggests that an individual's level of concern and perception of the problem may be a better predictor for multi-solution preference than how the problem was framed. Future research should focus on interventions that appeal to people's sense of severity and urgency towards certain problems, as well as how addressing this sense of severity and urgency benefits their well-being. Relevant studies have been done on risk perception. For example, one study found that a framing intervention highlighting specific threats to online privacy increased perceived severity and caused participants to take a more proactive role in guarding their personal information [54]. In other studies, severity framing highlighting known risks to patient and public health increased patient adherence to taking prescription drugs and following public health protocols [55, 56]. Future research on preference for multiple solutions can use similar interventions to highlight perceived severity and urgency to encourage action. Moreover, since risk perception is often associated with anxiety, further studies can examine how interventions on anxiety drive behavior change. For example, in some studies, participants with higher self-reported climate anxiety are more likely to engage in pro-environmental behavior [57].

In addition to perceived severity and urgency, our results suggest that reducing dichotomous thinking, or in other words, broadening thinking patterns, could assist in promoting preference for multiple solutions that may be more likely to succeed in addressing complex societal and environmental issues. However, a research gap exists on behavioral or educational interventions to encouraging broader thinking and reducing dichotomous thinking; previous studies focused on dichotomous thinking as a personality trait that is a predictor for borderline personality disorder [19, 58], eating disorders [59], and depression [19, 60]. Improving emotional well-being could be one way of reducing dichotomous thinking. For example, politically polarized individuals who often exhibit dichotomous tendencies, tend to report lower levels of emotional well-being [61]. Some interventions, such as reducing the time spent on social media and news can help reduce polarization, and therefore dichotomous thinking, as well as improving emotional well-being [62]. However, changing dichotomous thinking may ultimately require more than behavioral interventions, starting with changes in the education system to include a more multifaceted and diverse curriculum in schools [63, 64], improving science communication regarding complex and often counterintuitive issues [65], systemic changes to improve equality and inclusivity [12], as well as reducing political polarization [66]. These larger system-level changes could contribute to a less dichotomous sociopolitical environment where people are more open to different perspectives, emotionally balanced, and have less motivation to think dichotomously [67]. These changes are especially important in light the fact that some solutions (e.g., cash transfers to people experiencing homelessness) have been experimentally verified to be effective, yet are shunned by the public due to misconceptions regarding its efficacy or political affiliation that excludes certain solutions regardless of their efficacy.

A surprising exploratory finding in the current study was that the more conservative or more highly educated a participant was, the less likely they were to choose multi-solution

options. The fact that conservatives were less receptive to the multi-solution approach may be attributed to the fact that some of these societal problems can be seen as liberal problems. There is generally less support from conservative individuals on climate change or police reform, and conservatives are less likely to think that these problems require urgent action [68]. Previous studies showed that highly educated people tend to show a lower level of dichotomous thinking [10], which is the opposite of what we found. This result may be because the multi-solution option contained a smaller amount of information compared to the two binary options, which listed specific examples. This lack of information could be interpreted as poor information quality, making the multi-solution option less appealing to highly educated participants [69]. Second, while highly educated individuals are generally more comfortable with non-dichotomous thought [70, 71], they are also more likely to engage in more complex thinking that might not have been captured by the limited set of solutions in the current study. For example, a person can fully recognize the importance of both government and individual climate action, but still choose the former over the latter because they believe that government policies make it easier for individuals to engage in climate action, or vice versa [14]. Furthermore, we also did not explicitly ask the participants to consider the information presented in the frames when choosing their preferred solution. This could have caused them to rely more on their existing beliefs and preferences, which could explain why some demographic variables were good predictors of preferred solution.

In conclusion, the current study showed no demonstrable impact of framing on people's preference for multiple solutions to address societal problems. Future interventions should instead focus on helping people deal with the sense of severity and urgency they feel towards certain problems, or decreasing dichotomous thinking to encourage people to adopt multiple solutions to address complex environmental and social problems.

## Supporting information

**S1 File. Eight environmental and social problems presented in the four conditions.**
(DOCX)

**S2 File. Dichotomous thinking inventory.**
(DOCX)

**S3 File. Demographics.**
(DOCX)

**S4 File. Additional results.**
(DOCX)

## Author Contributions

**Conceptualization:** James Shyan-Tau Wu, Jiaying Zhao.

**Data curation:** James Shyan-Tau Wu.

**Formal analysis:** James Shyan-Tau Wu.

**Investigation:** James Shyan-Tau Wu, Jiaying Zhao.

**Methodology:** James Shyan-Tau Wu, Jiaying Zhao.

**Software:** James Shyan-Tau Wu.

**Supervision:** Claire Kremen, Jiaying Zhao.

**Writing – original draft:** James Shyan-Tau Wu.

**Writing – review & editing:** Claire Kremen, Jiaying Zhao.

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
