## [Decision Letter · Decision Letter 0]

16 Mar 2023

PONE-D-23-01648How does framing influence preference for multiple solutions to societal problems?PLOS ONE

Dear Dr. Wu,

Thank you for submitting your manuscript to PLOS ONE. After careful consideration, we feel that it has merit but does not fully meet PLOS ONE’s publication criteria as it currently stands. Therefore, we invite you to submit a revised version of the manuscript that addresses the points raised during the review process.

We look forward to receiving your revised manuscript.

Kind regards,

Denis Alves Coelho, PhD

Academic Editor

PLOS ONE

Journal Requirements:

Reviewers' comments:

Reviewer's Responses to Questions

**Comments to the Author**

1. Is the manuscript technically sound, and do the data support the conclusions?

Reviewer #1: Partly

Reviewer #2: Yes

2. Has the statistical analysis been performed appropriately and rigorously? 

Reviewer #1: Yes

Reviewer #2: Yes

3. Have the authors made all data underlying the findings in their manuscript fully available?

Reviewer #1: Yes

Reviewer #2: Yes

4. Is the manuscript presented in an intelligible fashion and written in standard English?

Reviewer #1: Yes

Reviewer #2: Yes

5. Review Comments to the Author

Reviewer #1: This study focuses on an important question, using an extensive sample, with preregistered research methods. The findings are important. I have several comments focusing especially on interpretation and the request for additional presentation of results.

(1) Interpretation A: Might the limited effect of framing compared with the control condition be partially attributed to the strong "manipulation check" that was used, where participants were required to place, in order, all of the presented bullet points per item. It is possible that this requirement may have elicited quite sustained and deep processing from all participants, in turn, encouraging a multi-solution perspective, and perhaps particularly so across the multiple items.

(2) Interpretation B: An additional potential reason for participants to choose the "multiple" solutions option (phrased as "Multiple solutions are needed in addition to 1 and 2") is that it is shorter than either the first or second options and could be chosen with a quick reading of the first and second options; if the first and second (longer) options seemed at least plausible on a quick reading, then choosing "both" involves less cognitive effort than carefully reading and differentiating between the first and second (longer) options. Thus part of the effects may be attributed to a tendency to prefer efficiency of decisions by participants. The "multiple solutions" response is also the same option across all of the items, and so (again) requires less careful reading/consideration.

Related to this: An exploratory analysis might examine if the same patterns are observed for only the first few items that participants responded to. Given that the order of stimulus presentation was randomized, this could be informative with regard to a fatigue or cognitive efficiency account.

(3) Analyses: Setting aside the number of participants who chose the multiple solutions option, I wonder if some (exploratory) suggestions might be derived from comparing the frequency of the choices for 1 vs. 2 for the different topics. For instance, whereas there seem to be generally approximately equal number of participants who chose option 1 vs. chose option 2 for the majority of topics, there is a marked difference for the topic of homelessness, with the proportion chosing binary option 1 nearly as high as those choosing multiple solutions. Also, given how different the responses to the homelessness topic are from the other topics, it might be worthwhile examining patterns excluding this topic.

(4) In the Introduction, with respect to framing in relation to psychological distance, it is notable that there are questions about the extent to which evidence in favor of construal level theory may be the outcome of publication bias; see, especially https://psyarxiv.com/r8nyu/

(5) The inverse relation between higher levels of education and choosing the multi-solution option seems puzzling, and one wonders if there are other possible accounts. The descriptive statistics for the demographic measures are not provided and should be given. Additionally, scatterplots of the inverse correlations should be examined and provided (e.g., in the supplementary material) to assess to what degree the (small) negative correlations are representative.

Specific Comments

- The manuscript does not have page numbers; please add.

- Introduction, first page, word missing, text should read, "allows humans to make decisions"

- Discussion, text amiss: "in some study, participants with higher..." (should read "studies"?)

Reviewer #2: Summary: This between-subjects experiment was designed to test whether describing real-world problems in different ways would have an influence on what people would choose as a "good solution" to the problem as well as how severe and how urgent they think the problem is. More specifically, the authors were interested in whether their different descriptions would result in participants being more likely to choose the option of "multiple solutions" as a "good solution" (and, potentially, less likely to choose only one specific solution as a "good solution." This is because their manipulation involved describing the problem and then listing multiple causes, multiple impacts, or multiple problems (three experimental conditions) as compared to listing multiple facts about the problem (one control condition). There were no significant differences found in the choices made by participants based on these differences in description. The impact of certain person variables--such as dichotomous thinking, political affiliation, and highest level of education--on these decisions were also explored. Interesting relationships were observed in exploratory analyses, such as the positive correlations between the likelihood of choosing the option of "multiple solutions" and perceiving the problem as more severe and more urgent and the negative correlation between the likelihood of choosing the option of "multiple solutions" and dichotomous thinking. My overall impression was generally positive. They make a strong argument for the importance of conducting more research on the topic of dichotomous thinking outside of clinical contexts, but I think there are some limitations that the authors need to address.

Major Issues:

1) The authors state at the beginning of their discussion section that their manipulation "had little impact" on the choices that people made. I understand that this is a common expression that should be interpreted as "had no demonstrable impact;" however, I think the latter more clearly describes the findings of this experiment. In the abstract, the authors state "these findings show a limited impact of framing on multi-solution preferences," and I find this even more misleading. The phrase "limited impact" is also used in their conclusion. None of the evidence presented suggests that the manipulation played a significant role in participants' decisions. I think this needs to be clearly summarized. At the same time, I acknowledge that the limitations of the study might have prevented the researchers from finding an effect that might be evidence under different conditions; however, that does not change their current results.

2) I noted a potential limitation that was not addressed by the authors. The author might not have found the effect they were interested in because they did not explicitly ask participants to consider the information provided when answering questions about the severity of the problem, the urgency of the problem, or their preferred solution. Without such an instruction participants may have relied more heavily on their pre-existing beliefs, which might help explain why some of the person variable were good predictors of whether people would choose the option of "multiple solutions."

Minor Issues:

1) I had an issue with the question that was used to ask participants about their preferred solution. The question asks the participants to select a "good solution;" however, I was not clear on why the question was not about the "best solution" and whether such a change might have made a difference.

2) I could not tell which style guide was being used for the reference section. There were some inconsistencies in how names were formatted and whether periods were or were not used following authors' initials.

6. PLOS authors have the option to publish the peer review history of their article (what does this mean?). If published, this will include your full peer review and any attached files.

Reviewer #1: No

Reviewer #2: No

---

## [Author Response · Author response to Decision Letter 0]

9 Apr 2023

Please see attachment (Response to Reviewers). Thank you.

---

## [Decision Letter · Decision Letter 1]

2 May 2023

How does framing influence preference for multiple solutions to societal problems?

PONE-D-23-01648R1

Dear Dr. Wu,

We’re pleased to inform you that your manuscript has been judged scientifically suitable for publication and will be formally accepted for publication once it meets all outstanding technical requirements.

Kind regards,

Denis Alves Coelho, PhD

Academic Editor

PLOS ONE

Reviewers' comments:

Reviewer's Responses to Questions

**Comments to the Author**

1. If the authors have adequately addressed your comments raised in a previous round of review and you feel that this manuscript is now acceptable for publication, you may indicate that here to bypass the “Comments to the Author” section, enter your conflict of interest statement in the “Confidential to Editor” section, and submit your "Accept" recommendation.

Reviewer #1: All comments have been addressed

Reviewer #2: All comments have been addressed

2. Is the manuscript technically sound, and do the data support the conclusions?

Reviewer #1: Yes

Reviewer #2: Yes

3. Has the statistical analysis been performed appropriately and rigorously? 

Reviewer #1: Yes

Reviewer #2: Yes

4. Have the authors made all data underlying the findings in their manuscript fully available?

Reviewer #1: Yes

Reviewer #2: Yes

5. Is the manuscript presented in an intelligible fashion and written in standard English?

Reviewer #1: Yes

Reviewer #2: Yes

6. Review Comments to the Author

Reviewer #1: There are a few places with missing words, or minor typos, such as:

p. 15: how addressing this sense of severity and urgency and benefits their well-being.

p. 17: This result may be because the multi-solution option containing a smaller amount of information

p. 6 Knowing this, multi-cause, multi-impact, or multi-solution framings might be employed encourage

Reviewer #2: Thank you for addressing my comments. The only issues I noted had to do with potential typos or issues with specific words or phrases.

Page 4 line 10: "Many of past" should be "Many past."

Page 14 line 5: I believe that "are" should be "is" because it refers to "employing."

Page 15 line 2: I am not sure what is meant by "that resort to psychological distance." Are you referring to framing interventions that focus on or rely on psychological distance? If so, that is not how I am interpreting the current phrasing.

Page 16 lines 18-19: The phrase "have been demonstrated to be experimentally verified to be effective" could be simplified (e.g., "have been experimentally verified to be effective").

7. PLOS authors have the option to publish the peer review history of their article (what does this mean?). If published, this will include your full peer review and any attached files.

Reviewer #1: No

Reviewer #2: No

---

## [Editor Report · Acceptance letter]

8 May 2023

PONE-D-23-01648R1 

How does framing influence preference for multiple solutions to societal problems? 

Dear Dr. Wu:

I'm pleased to inform you that your manuscript has been deemed suitable for publication in PLOS ONE. Congratulations! Your manuscript is now with our production department. 

Kind regards, 

on behalf of

Dr. Denis Alves Coelho 

Academic Editor

PLOS ONE